# Effects of Mullite, Maghemite, and Silver Nanoparticles Incorporated in β-Wollastonite on Tensile Strength, Magnetism, Bioactivity, and Antimicrobial Activity

**DOI:** 10.3390/ma14164643

**Published:** 2021-08-18

**Authors:** Hamisah Ismail, Farah ‘Atiqah Abdul Azam, Zalita Zainuddin, Hamidun Bunawan, Muhamad Afiq Akbar, Hasmaliza Mohamad, Muhammad Azmi Abdul Hamid

**Affiliations:** 1School of Applied Physics, Faculty of Science & Technology, Universiti Kebangsaan Malaysia, Bangi 43600, Selangor, Malaysia; hamisahismail@usm.my (H.I.); qfaara@gmail.com (F.‘A.A.A.); zazai@ukm.edu.my (Z.Z.); 2School of Materials and Mineral Resources Engineering, Universiti Sains Malaysia, Nibong Tebal 14300, Penang, Malaysia; hasmaliza@usm.my; 3Institute of Systems Biology, Universiti Kebangsaan Malaysia, Bangi 43600, Selangor, Malaysia; hamidun.bunawan@ukm.edu.my (H.B.); muhdafiq.akbar@gmail.com (M.A.A.)

**Keywords:** β-wollastonite, mullite, maghemite, silver, bioactivity

## Abstract

β-wollastonite (βW) has sparked much interest in bone defect recovery and regeneration. Biomaterial-associated infections and reactions between implants with human cells have become a standard clinical concern. In this study, a green synthesized βW, synthesized from rice husk ash and a calcined limestone precursor, was incorporated with mullite, maghemite, and silver to produce β wollastonite composite (βWMAF) to enhance the tensile strength and antibacterial properties. The addition of mullite to the βWMAF increased the tensile strength compared to βW. In vitro bioactivity, antibacterial efficacy, and physicochemical properties of the β-wollastonite and βWMAF were characterized. βW and βWMAF samples formed apatite spherules when immersed in simulated body fluid (SBF) for 1 day. In conclusion, βWMAF, according to the tensile strength, bioactivity, and antibacterial activity, was observed in this research and appropriate for the reconstruction of cancellous bone defects.

## 1. Introduction

At this time, many commercial bone replacements are being used to replace bone implants in orthopedic trauma surgery procedures. However, their widespread application is limited because they do not meet several of the regenerative needs of bone tissue and practical needs related to handling and surgery. There are potential and growing applications of bioactive calcium silicate for commercial bone replacement, such as in dentistry [1], load-bearing [2], and scaffold [3,4], but it has some limitations, such as high brittleness [5], low antibacterial resistance [6], and no magnetic properties [7]. Due to the problems encountered, a dopant material should be added to the bioactive calcium silicate to improve the implant’s properties. In the present study, examples of dopant materials were applied in calcium silicate ceramics, such as strontium (Sr) [8], copper (Cu) [9], magnesium (Mg) [10], zirconia (ZrO_2_) [10], and titania (TiO_2_) [11]. The justification for using these particular metal oxides for doping is that they contain metallic elements that are similar to trace elements found in bone, such as zinc (Zn), magnesium (Mg), and strontium (Sr), all of which have been shown to have potential benefits in encouraging formation bone growth [12]. The use of zirconia and titanium alloys as implantable orthopedic materials in the past refers to the doping of other metal oxides containing zirconium (Zr) [13] and titanium (Ti) [14]. Using metal oxide nanoparticles as a dopant improved antibacterial activity while minimizing toxicity, chemical stability, and thermal resistance.

Iron compound nanoparticles or magnetic nanoparticles have an essential aspect in biomedical engineering and biology. Magnetite (Fe_3_O_4_) and maghemite (γ-Fe_2_O_3_), the two typical forms of magnetic nanoparticles, are most suitable in biomedical science and engineering due to their non-toxic property, biocompatibility, high surface to volume ratio, ability to excellently carry a broad range of biological ligands, and existence of natural routes for its degradation [15,16], due to a superior class of metal oxide nanoparticles with unique magnetic properties and more outstanding biocompatibility. Laterally, with the growing uses of magnetic nanoparticles, magnetic nanoparticles’ possible toxic effects have been a far-reaching burden [17,18,19,20]. Numerous outcomes point out that magnetic nanoparticles suggestively diminish the cell viability of human mesothelioma [21], epithelial cell lines [22], human macrophage [22], and hinder the standard formation of PC12 neuronal cell morphology [23]. In other studies, the dimercaptosuccinic acid (DMSA)-coated MNPs reduce the mitochondrial activity of human fibroblasts at a high concentration of magnetic nanoparticles [17]. However, magnetic nanoparticles’ cytotoxicity statistics are challenging since magnetic nanoparticles’ toxic effects are influenced by several factors, such as magnetic properties, distribution of size, and surface coating [17]. Therefore, it is critical to select the cell line for the cytotoxicity assessment of precise magnetic nanoparticles. However, the cytotoxicity studies of magnetic nanoparticles incorporated in calcium silicate bioceramics are restricted because the cytotoxicity mechanism is still in early investigation [18].

Currently, antimicrobial activities of the silver ion have been reported against both Gram-positive and Gram-negative bacteria, protozoa, fungi, and certain viruses [19]. Silver ions are widely used to inhibit bacterial growth in various medical utilization, including catheters, dental biomaterials, burn trauma, and surgical appliances [9,19,20,24]. Some researchers have described the deliberate addition of chemical components into biomaterials to gain antimicrobial effects [25,26], and silver is the most generally used because of its excellent antibacterial activity [27,28]. Commonly, silver nanoparticles are less toxic than silver ions [28] and other metals [29]. The doping of the silver, copper, and zinc with hydroxyapatite, which exhibited potential action on *Escherichia coli* (*E. coli*) and *Staphylococcus aureus* (*S. aureus*) microbial growth inhibition, has been widely studied [30,31,32]. Meanwhile, the consumption of doping antibacterial agents, such as silver nanoparticles [33,34] and copper [35,36] doped in β-wollastonite, is still new to explore. Lohbauer et al. mentioned that approximately one ppm with a consistent release of silver ions has no cytotoxic effect on the viability of human epidermal keratinocytes [37]. Another researcher reported that nanocrystal composite of HA doped Ag exhibited biocompatible or not toxic effects on L929 cells when an up to 2.2 ppm concentration of 50/1000 μg/mL was applied. Otherwise, for HA/Ag at 197 ppm, cell viability reduced to 40%, showing increased toxicity [38].

This research aims to create a green synthesis of β-wollastonite obtained from rice husk ash and calcined limestone incorporated with mullite, maghemite, and silver nanoparticles to test its tensile strength, magnetic properties, bioactivity, and antibacterial potential.

## 2. Materials and Methods

The β-wollastonite (βW) was synthesized from agricultural and natural rice husk ash and limestone sources. The silica (SiO_2_) source from fired rice husk was obtained from a rice mill industry in Penang, Malaysia. Meanwhile, calcium oxide (CaO) was collected from a nearby limestone quarry in Perak, Malaysia. The β-wollastonite composite (βWMAF) consists of 20 wt.% of mullite, 5 wt.% of maghemite (γ-Fe_2_O_3_), and 5 wt.% silver (Ag). Mullite was obtained from the calcination of andalusite for 8 h at 1450 °C and collected from a mining quarry in Terengganu, Malaysia. Mullite selection of 20 wt.% refers to a previous study by Farah et al. [39], which found optimum compressive strength was obtained when using 20 wt.%. Silver nanoparticles with a purity of 99% and a particle size of <100 nm were purchased from Sigma-Aldrich (Petaling Jaya, Malaysia). The results were taken from a previous study by Farah et al. [34], which showed optimum results for antibacterial properties when 5 wt.% was used in the previous study. Gram-positive *Staphylococcus aureus* (*S. aureus*) ATCC11632 and Gram-negative *Escherichia coli* (*E. coli*) ATCC10536 were used for antibacterial activity evaluation in the antibacterial study. A culture agar (nutrient agar 17 g/L from LB, Wako) and a culture broth (Mueller Hinton Broth 21 6/L from LB, Wako) were used as a media cultivation nutrient. The maghemite nanoparticle was self-synthesized using the co-precipitation technique, and the raw materials were used as anhydrous iron (II) chloride (FeCl_2_, 98% purity) and anhydrous iron (III) chloride (FeCl_3_, 97% purity), supplied by Sigma-Aldrich (Petaling Jaya, Malaysia.

### 2.1. Preparation of βW and βWMAF Disc

A solid-state reaction method assisted by using an autoclave reactor was adopted to prepare β-wollastonite (βW); the CaO:SiO_2_ is 55:45. The CaO and SiO_2_ were mixed using deionized water, autoclaved for 8 h at 135 °C, and kept to dry in the oven for 24 h. Afterward, the dried mixture was sintered at 950 °C for 2 h. A brief explanation of the preparation of βW was completed and described in the previous study [40,41].

To produce the βWMAF sample, the mullite powder, maghemite, and silver were milled using ball milling at 250 rpm for 15 min to obtain the homogenous mixture. The βW as a control sample and βWMAF mixture were pressed under 2.5 tons to get a disc (12.0 ± 0.1 mm diameter × 2.0 ± 0.1 mm thickness).

### 2.2. Characterization of βW and βWMAF

The phase identification of βW, mullite, maghemite, silver, and βWMAF was detected via X-ray diffraction (XRD, D8 Advance, Bruker, Karlsruhe, Germany) using Cu Kα radiation (λ = 0.15406 nm) with a step size of 0.025°/step, a counting time of 0.1 s, and the Bragg’s angle ranging from 20–70°. The morphology and elemental analyses were examined using a field emission scanning electron microscope (FESEM, Carl Zeiss, Compact, Jena, Germany) and an energy dispersive (EDS, IncaEnergy, Oxfordshire, UK) spectrometer, and the samples were platinum sputter-coated before analysis. The microscope was operated at a voltage of 3.0 kV.

### 2.3. Diametral Tensile Strength βW and βWMAF

The diametral compression or Brazilian disc test is an excellent way to measure the tensile strength of brittle materials, such as ceramic and biomaterials [42]. Theory of elasticity, finite element analysis, and photoelasticity studies have all confirmed the state of stress. The resulting findings were curiously consistent and in perfect agreement with Hertz’s classical analysis [43]. Except for the immediate vicinity of the loading points, there was a convergence of the stress solutions throughout the disc in all cases. The diametral compression test yielded strength values that were sometimes comparable to flexure strengths but not always [44].

In this experiment, the diametral tensile strength of the βW and βWMAF was evaluated using the Brazilian test, referring to the ASTM C496 standard [39]. The length and diameter of five specimen discs (12.0 ± 0.1 mm in diameter × 2.0 ± 0.1 mm in thickness) were measured. The sample was positioned between steel platens and tested on a Universal Testing Machine (Instron 8874, Instron, Norwood, MA, United States) with the load cell 5 kN and a loading rate of 0.1 mm/min [39]. The maximum compression load at failure was calculated from the load-deflection curves identified. The tensile stress here is directly proportional to the applied compressive load. Using the following relationship, the strength value of the βW and βWMAF samples was determined: diametral tensile strength, σ_t_ = 2P/πDT, where P is the peak load (N), D is the diameter (mm), and T is the sample thickness (mm).

### 2.4. Magnetic Properties

Magnetic properties were evaluated using a vibrating sample magnetometer (VSM, Lakeshore 7410, Lake Shore Cryotronics, Inc., Westerville, OH, United States with an applied magnetic field of 14 k Oe at room temperature for maghemite (γ-Fe2O3) βW and βWMAF powders. The external magnetic field vibration could be applied both in the transverse and longitudinal directions of the vibration. Each sample’s magnetic moment was measured over a range of magnetic fields applied between −1.4 × 10^3^ and 1.4 × 10^3^ Oe with a 0.1 emu sensitivity.

### 2.5. In-Vitro Bioactivity

The ability to form apatite on the surface of the βW and βWMAF discs was tested using an SBF solution whose ionic composition is similar to that of human physiological body fluid. Generally, βW and βWMAF discs were soaked in 30 mL of SBF solution for 1, 7, and 14 days in a polyethylene bottle and retained in an incubator at 37 °C, referring to the procedure by Kokubo and Takadama [45]. Every three days, the SBF solution was replaced. After soaking each day, the βW and βWMAF discs were removed from the SBF solution, immersed in acetone for 2 h, washed three times with deionized water, and dried in the incubator. The SBF solutions’ pH values were measured after 1, 7, and 14 days of soaking βW and βWMAF discs. Phase, functional group, morphology observation, and elemental composition were used to determine the apatite formation on βW and βWMAF disc surfaces.

### 2.6. Ca, P, Fe, and Ag Release in SBF Solution

To evaluate the ion dissolution from βW and βWMAF discs, the SBF was prepared according to Kokubo’s method [45]. The βW and β-WMAF discs were put in a 30 mL polyethylene bottle immersed in SBF solutions at 36.5 °C in an incubator for 1, 7, and 14 days without changing the solution. The concentrations of Si, Ca, P, Fe, and Ag ions in the SBF solutions were calculated by an inductively coupled plasma mass spectrometer (ICP-MS, Perkin Elmer, Waltham, MA, United State).

### 2.7. Antimicrobial Activity

Using *Escherichia coli* ATCC10536 and *Staphylococcus aureus* ATCC11632, the antimicrobial activity was carried out using a disc diffusion method using Mueller Hinton Agar (MHA). All the specimens and apparatus were sterilized in an autoclave at 121 °C for 15 min), and all the tasks were inside the laminar airflow chamber. The bacteria cultures were prepared in nutrient broth (Difco 234000, Nelson-Jameson, Marshfield, WI, USA) at a concentration of~1.5 × 10^8^ CFU/mL corresponding to the density of 0.5 MacFarland. Next, under sterilized conditions, *E. coli* and *S. aureus* cultures were swabbed on the agar petri dish using a sterile cotton swab. The βWMAF disc was placed on the agar, followed by the βW disc as negative control and Ag disk as a positive control. These petri discs were incubated at 37 °C for 24 h. Subsequently, the zone of inhibition (ZOI) was calculated in millimeters (mm) for duplicate specimens for each sample.

### 2.8. Statistical Analyses

The bioactivity test, magnetization test, antibacterial test, and five samples for diametral tensile strength were duplicated. For n = 2 and 5, all data were expressed as means and standard deviations.

## 3. Results and Discussion

### 3.1. Characterization of βW and βWMAF

Figure 1 exhibits XRD analysis of all the raw materials: (a) maghemite (γ-Fe_2_O_3_), (b) mullite, (c) silver, (d) βW, and (e) βWMAF powders. The prominent peak of each raw material was an exhibit at βWMAF (Figure 1e). The prominent peak of maghemite (γ-Fe_2_O_3_) phase was observed at theta 35°, for mullite at 26.1°, silver (Ag) clearly at 38.2°, and most of the peak of βWMAF was dominant by β-wollastonite phase (β-CaSiO_3_) at 30°, 50.2°, 53.3°, and 39.1°. Referring to previous reports, amorphous CaSiO_3_ has a faster apatite-formation capacity owing to the accelerated release of Ca ions relative to alpha-wollastonite (α-CaSiO_3_) and beta-wollastonite (β-CaSiO_3_) [46]. Consequently, βWMAF materials may enhance bioactivity. For reference, the patterns of maghemite (ICDD no: 39-1346), mullite (ICDD no: 74-4144), silver (ICDD no: 04-0783), and β-wollastonite (ICDD no: 043-1460) are referred.

The micrograph of the βW and βWMAF discs are presented in Figure 2 and act as a control sample (unsoaked) for bioactivity test. Morphology observation showed that the βW and βWMAF discs show a porous structure (Figure 2a,b). The βW disc image showed that the surface was smooth and exhibited a uniform coral-like shape (Figure 2a), and the EDX spectrum displayed the presence of Ca, Si, O, and P, where the P came from rice husk ash. The surface disc of βWMAF showed a bulky, irregular shape and uneven surface due to many elements, such as mullite, silver, and maghemite, and it was confirmed by the EDX spectrum, which showed the presence of Ca, Si, O, Fe, Ag, Al, and P elements (Figure 2b). Another reason was that the milling process used to prepare the βWMAF mixture may have changed the coral-like microstructure of βW into a bulky, irregular shape with an uneven surface (Figure 2b). The silver, maghemite, and mullite particles are scattered throughout the matrix, with some of them clustered. One of the most critical aspects of successful composite preparation is the uniform distribution of silver, maghemite, and mullite particles.

### 3.2. Diametral Tensile Strength (DTS) βW and βWMAF

The addition of mullite to the βWMAF increased the DTS compared to βW. The maximum strength of the βW sample in the absence of reinforcement was 0.69 ± 0.13 MPa. The addition of 20 wt.% mullite as reinforcement to the βWMAF increased the strength to 2.39 ± 0.25 MPa, and it was appropriate for cancellous bone implant or scaffold with a compressive strength between 2–12 MPa [47]. Farah et al. obtained the highest DTS value (4.4 ± 0.15 MPa) when using 20 wt.% and the lowest DTS value (2.9 ± 0.15 MPa) when adding 30 wt.% to pseudowollastonite sintered at 1000 °C [39]. Zhang et al. also reported the effects of adding various alumina (0.25, 2, and 5 wt.%) on raising the mechanical properties and low-temperature degradation of yttria-stabilized TZP ceramics for dental implants [41]. As a result, the presence of alumina particles at a concentration of 15% by weight in calcium silicate matrix strength increased the fracture toughness and hardness of calcium silicate [40].

### 3.3. Magnetic Properties

Magnetization curves of the maghemite (γ-Fe_2_O_3_), β-wollastonite (βW), and βWMAF powders at room temperature are shown in Figure 3. The βW materials had no magnetization and showed a diamagnetic property due to their non-magnetic nature. No hysteresis loop for the βW, due to the B–H relationship, was linear. Simultaneously, the βWMAF had a magnetization of 1.731 emu g^−1^ at 25 kOe, because the combined Fe ions formed a magnetite structure. Similarly, almost no hysteresis loops were found on the curves of the βWMAF, indicating superparamagnetic behavior. Further research needs to investigate βWMAF properties due to the heating ability, specifically, using βWMAF as thermoseeds for hyperthermia treatment of solid cancer tumors under an alternating magnetic field.

### 3.4. In Vitro Bioactivity

Figure 4 shows the XRD diffraction patterns of the βW and βWMAF samples after 14 days of soaking in SBF. The XRD patterns of βW and βWMAF samples before and after soaking showed a sharp peak at 2θ = 30.0°, corresponding to the beta-wollastonite (β-CaSiO_3_) phase (ICDD no. 00-043-1460), as shown in Figure 4a,b. The resulting XRD trends revealed that the intensity of the β-wollastonite phase in the βW control was reduced after 1 day of soaking, and the calcite phase (CaO) (ICCCD no. 00-05-586) was present at 29.3 °C (Figure 4a). The peak intensity of the formed calcite increased dramatically from day 1 to day 7 of soaking and unfortunately disappeared at day 14; this could be attributed to the deposition of amorphous precipitate on the surface of βW. The amorphous precipitate refers to the peak of the hydroxyapatite (HA, ICDD no. 00-09-0302). It was observed as early as day 7; the peak became more apparent on day 14 of the soaking process. However, the intensity for β-wollastonite was still high compared to HA after 14 days of soaking.

Meanwhile, for βWMAF, some phases were detected, such as β-wollastonite, mullite, maghemite, and silver, before soaking in the SBF for 14 days (Figure 4b). The β-wollastonite peak intensity at 2θ = 30.0° was decreased and surprisingly disappeared on day 7 of soaking. The calcite peak at 2θ = 29.3° was present after a day of soaking until day 14. Hydroxyapatite (HA) peak was discernible at day 7 of soaking in the SBF compared to the βW sample (Figure 4a). Unpredictably, the intensity peak for calcite and HA was quite the same, and the rest peak of mullite, maghemite, and silver was still there after 14 days of soaking (Figure 4b).

The calcium (Ca^2+^) and silicon (Si^4+^) ions released by the samples entered the SBF solution as they were immersed in it. The H+ ions replaced the Ca^2+^ ions in the samples in the SBF solution as the samples were immersed in the solution, resulting in the formation of silanol (Si–OH) on the surface and an increase in the pH of the SBF solution [48]. The silanol (Si–OH) layer formed on the sample surface aided in forming the hydroxyapatite (HA) phase. The SBF’s phosphate (PO_4_^3−^) ions reacted with Ca^2+^ ions in the solution to form the HA phase on the sample [49]. As stated by Sainz et al. [50], this Equation (1) is comparable to the common equation that exists during the HA formation process (Ca_10_(PO_4_)_6_(OH)_2_):10Ca^2+^ + 6PO_4_^3−^ + 2OH^−^ ⇄ Ca_10_(PO_4_)_6_(OH)_2_(1)

According to the XRD patterns, replacing SBF every three days had a significant impact on the formation of apatite phases on the surface of the βW and βWMAF samples because there was an adequate supply of PO_4_^3^^−^ ions from the SBF solution to be mixed with Ca^2+^ ions from the samples to form the apatite layer. In summary, the βWMAF sample was more bioactive than βW, where no β-wollastonite peak was detected up to 14 days of soaking. In this study, apatite formation on the surface of βWMAF was influenced by maghemite and silver. Some researchers also confirmed the use of silver and maghemite as dopant materials [46,51,52]. Meanwhile, mullite does not help in apatite formation due to its inert properties in nature [51].

Figure 5 depicts the FTIR spectra of βW and βWMAF samples before and after soaking for up to 14 days. As other researchers have discovered, an extensive OH- absorption band at 3600 and 2700 cm^−1^ and a small water absorption band at 1600 cm^−1^ can be perceived in these spectra for both samples [35,50]. Before soaking, the absorption band characteristics of βW and βWMAF demonstrated the formation of a β-wollastonite phase dependent on the silicone ion bending bands (Si–O) at 1008.50 and 930.76 cm^−1^ [40,52]. β-wollastonite phases characterized silicon ion stretching bands (Si–O–Si) at 897.78 and 898.90 cm^−1^. Carbonate CO_3_^2-^ infrared (IR) absorption of βW and βWMAF samples were responsible for the 1419.23 and 1420.03 cm^−1^ bands. Previous research has shown that the bioactive process began between the 1008.50 and 930.76 cm^−1^ bands, as evidenced by the presence of non-bridging oxygen-stretching modes [1,29]. These bands regulate the rate of silicate matrix formation, which resulted in the formation of the silanol group at the surface of the βW and βWMAF. The same reaction was observed in the sample soaked for 1 day, as evidenced by the Si ions’ decreased strength. This situation indicated the need for Si ions to grow the Si–OH group, which induced apatite formation from day 1 onwards [1]. After soaking for 7 days, the P–O bending band’s deep width was observed to be 1021.61 cm^−1^ for the βWMAF sample, higher than the βW sample, which also occurred in stoichiometric apatite [31]. This finding was supported by XRD data, which showed that βWMAF had a higher apatite formation peak than βW (Figure 4).

Figure 6 depicts the FESEM/EDX analyses of surface apatite after up to 14 days of soaking in SBF for βW and βWMAF. As visualized for day 1, spherical precipitates were observable on the surface of the βW and βWMAF. Similarly, both samples also developed spherules of about 1 μm in size, showing an apparent inclination to attract apatite precipitates to their surfaces. Up to 7 days of soaking, crystal growth on the spherules surface became more significant for both samples, as shown in Figure 6. Next, for 14 days of soaking, spherules were incorporated to become more significant, and the spherules’ shape disappeared in both samples. From day 1 to day 14, soaked samples were SEM/EDX analyzed to confirm that the identified precipitates were apatite formed from an SBF solution. Colfen [50] introduced the surface-directed calcium phosphate mineralization process and observed the improvements in crystal structures in the βW and βWMAF samples that have been soaked for long periods. After soaking the βW and βWMAF samples in SBF, the mechanism began by creating ions such as Si^4+^, Ca^2+^, Fe^3+^, Ag^+^, H^+^, OH^−^, and HPO_4_^2−^. A day after soaking, the pre-nucleation aggregates in the arrangement’s spherule shape were in balance with the ions in the SBF solution, as revealed in Figure 6a,b; βW and βWMAF samples, respectively. During soaking, pre-nucleation aggregates began to form on the spherules near the surface of the βW and βWMAF, with loose aggregates still in the solution. There was more aggregation, inducing densification at the surface of the spherules of βW and βWMAF. At the surface of the βW and βWMAF, the nucleation of amorphous spherule particles started to occur, and these particles emerged as amorphous calcium phosphate (ACP) on the βW and βWMAF surfaces, respectively (Figure 6a,b). Next, crystallization occurred in the amorphous spherule region, directed by the surface, in the mineralization of calcium phosphate. The Ca/P ratios of the βW and βWMAF immersed surfaces varied from 6.4 to 2.25 and 9.4 to 1.76, respectively, as seen in Figure 7a. Interestingly, the Ca/P ratio decreased significantly within the presence of maghemite and silver due to the variety of ions dissolution, such as Al^3+^, Fe^3+^, and Ag^+^ in βWMAF. The factor may cause the crystal growth on the spherules surface Ca/P ratio, which was 2.25 (βW) and 1.76 (βWMAF) upon 14 days of immersing (Figure 7a). This value met the amorphous calcium phosphate (ACP) Ca/P ratio, varying from 1.2 to 2.2 [53].

The pH study of βW and βWMAF disc samples suggested an ion exchange process between the sample surface and SBF solution. Changes in the pH of samples in SBF are presented in Figure 7b. A quick increase in the pH solution showed the exchange of ions between βW and βWMAF samples, such as Ca^2+^, Si^4+^, Ag^+^, Fe^2+^, Al^3+^, with ions from SBF solution, such as OH^−^, PO_4_^3−^, and Na^+^. The pH value for βW samples from day 1 soaking was 7.4 up to 8.10 and decreased to 7.60 and, for the βWMAF sample, from 7.4 up to 8.43 and decreased to 7.62 after 14 days of soaking. The pH of the solution changed in a small range for βW and βWMAF samples, which indicated that samples were chemically stable in SBF solution and did not show any acidic nature in SBF. Instead, ion release would also alter biomaterial pH, influencing cell growth and osseointegration [54].

### 3.5. Ca, P, Fe, and Ag Release in SBF Solution

Figure 8 shows the release of Ca, Si, P, Fe, and Ag ions for 1, 7, and 14 days in the SBF solution being calculated using ICP-MS. Therefore, the release of Ca, Si, P, Fe, and Ag ions from βWMAF in the SBF solution was examined. As shown in Figure 9, except for the P ion, the drastic release of increased Ca, Si, Fe, and Ag ions was observed, with the rise in immersing time decreased. Parallel Ca and Si ion concentrations increased with increasing Fe substitution, which may be attributable to Fe ions having more Si and O network bonds than Ca ions, leading to a more robust βWMAF network. After the next 7 days, Ca, Si, Fe, and Ag’s dissolution rate slowed until day 14. After 14 days, the total Ca, Si, Fe, Ag, and P in the βWMAF were 265.81, 74.06, 1.18, 2.87, and 0.033 ppm, respectively.

In comparison, the Ag ion release peaked around days 2–6, and the decrease of Ag ion was subsequently observed after day 7. Ag ions’ complete release during the initial process could provide successful antibacterial activity, avoiding infections linked to the surgical site. Ag’s minimum bacterial concentration is 0.1 ppm, so this total of released Ag ions was sufficiently large to prevent bacteria growth.

### 3.6. Antimicrobial Study

In order to investigate the antimicrobial activity of βWMAF (12 mm diameter disc), the disc diffusion method was applied by using *E. coli* and *S. aureus* bacteria as test culture. The antibacterial activity can be observed from the clear zone of bacterial inhibition area around the test disk, as seen in Figure 9. Figure 10 shows the zone of inhibition (ZOI) measured by the disc diffusion test as a function of Ag for a βW and βWMAF disc for *E. coli* and *S. aureus*. A surprisingly good inhibition zone was observed for βWMAF on two different bacteria (Figure 9). The inhibition zone (ZOI) for *E. coli* was estimated at 14.1 mm for βWMAF and Ag, as seen in Figure 10. For *S. aureus*, Ag showed a higher ZOI than βWMAF, with readings of 17.0 mm and 15.8 mm, respectively, as described in Figure 10. The observations also found that pathogenic inhibition was seen less in *S. aureus* than in *E. coli* because of a generous thicker peptidoglycan layer, being characteristic of *S. aureus*. Several studies showed that *S. aureus* is more resistant to antimicrobial compounds than *E. coli* [55].

However, no inhibition zones were found on βW, implying that βW lacked bactericidal or bacteriostatic activity. The most common cause of implantation infections that resulted in implant failure was bacterial adhesion to the implant material. Several studies on metal ion-based modification by loading antibacterial agents such as Cu, Ag, and Zn were published to improve silicate bioceramic antimicrobial activity [56]. This ion-based alteration could also influence the toxicity and degree of structural integrity. Therefore, the currently prepared βW using calcined limestone and rice husk ash demonstrated possible antibacterial agent doping on microbial growth inhibition, close to other researchers’ findings [57,58]. The mechanism for the antimicrobial action of silver ions is not adequately understood. However, the prepared βWMAF may also be a possible candidate for biomedical applications.

## 4. Conclusions

The magnetism, bioactivity, and antibacterial activity of βW and βWMAF synthesized from locally available materials, such as rice husk ash (RHA) and limestone ash, were investigated. The inclusion of 20 wt.% mullite into the βW increased the compressive strength up to 200%, which is appropriate for the cancellous bone application; moreover, adding 5 wt.% Ag into the βW produced an excellent antibacterial property against *E. coli* and *S. aureus*. In vitro bioactivity findings revealed that apatite crystals were close-packed and fragile, with a quicker growth rate on βWMAF than on βW. The most crucial antimicrobial analysis indicated that βWMAF samples were antibacterial against *E. coli* and *S. aureus*. Furthermore, cytotoxic assays of the βW and βWMAF must be critically characterized. Another issue is creating a βWMAF bioceramic composite with strong antibacterial characteristics and low cytotoxic risk. As a result, further changes to the fabrication to improve crystallinity and reduce the dissolution of Fe and Ag during the early stage may render the bioceramics composite non-toxic to human cells. As a result, the current study findings provide new insights into the first step of future complex work to fabricate magnetic antibacterial β-wollastonite bioceramic composite scaffolds for bone regeneration. Development of calcium and silica-based material biowaste further improves the significant advantages in biomaterial applications.

## Figures and Tables

**Figure 1 materials-14-04643-f001:**
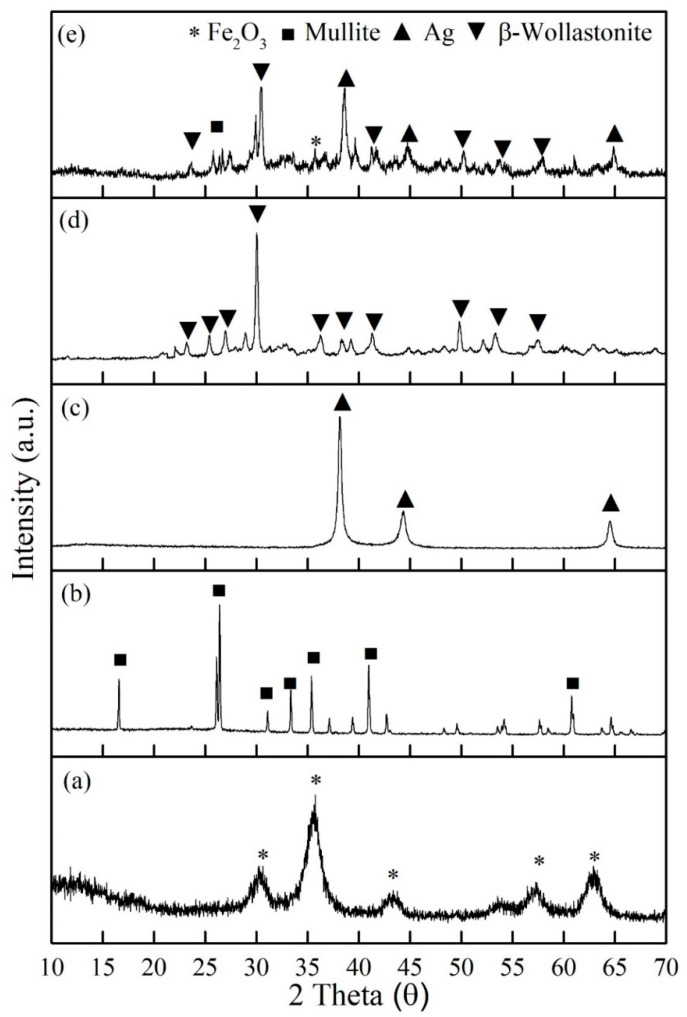
XRD analysis of all the raw materials; (**a**) maghemite (γ-Fe_2_O_3_), (**b**) mullite, (**c**) silver, (**d**) βW, and (**e**) βWMAF composite powders.

**Figure 2 materials-14-04643-f002:**
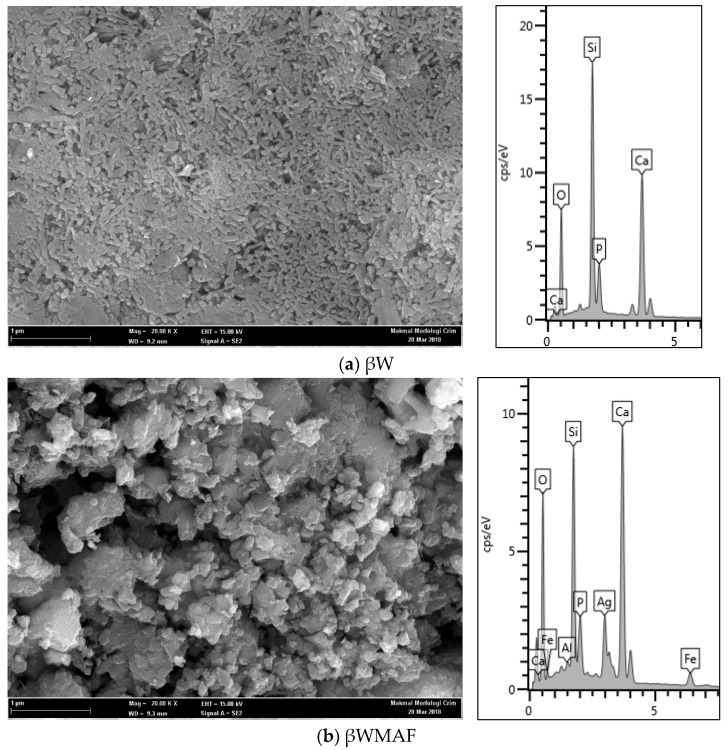
Morphology observation and EDAX spectra of (**a**) βW and (**b**) βWMAF discs.

**Figure 3 materials-14-04643-f003:**
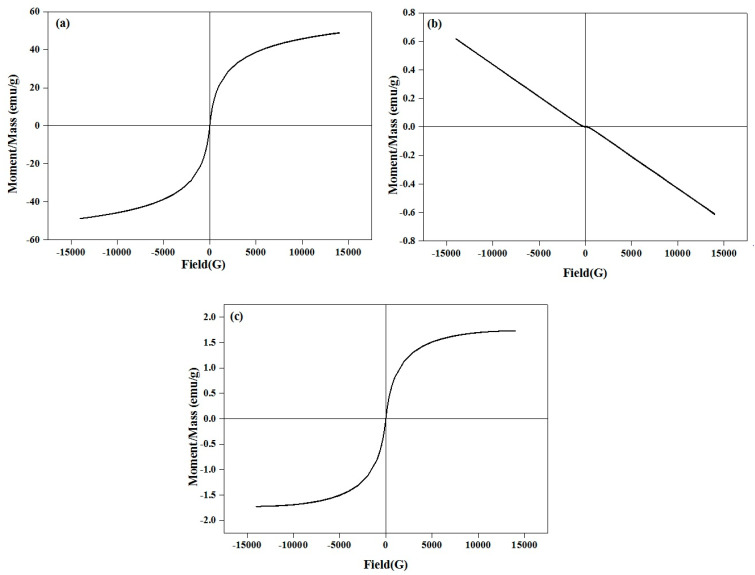
Hysteresis loop of the (**a**) maghemite (γ-Fe_2_O_3_), (**b**) βW, and (**c**) βWMAF powders.

**Figure 4 materials-14-04643-f004:**
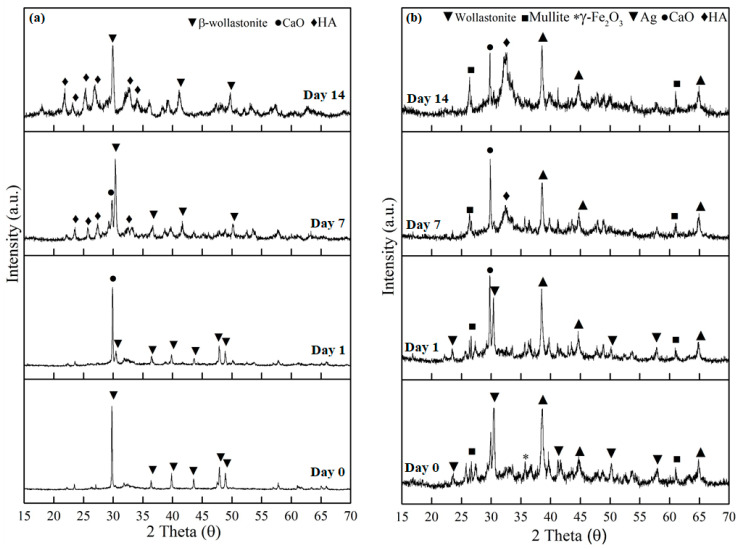
The XRD patterns of the (**a**) βW and (**b**) βWMAF before and after soaking until day 14 in an SBF solution.

**Figure 5 materials-14-04643-f005:**
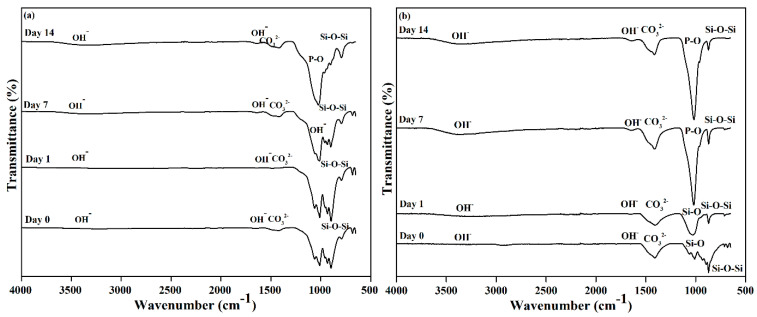
FTIR spectra of the (**a**) βW and (**b**) βWMAF before and after soaking in SBF from day 1 until day 14.

**Figure 6 materials-14-04643-f006:**
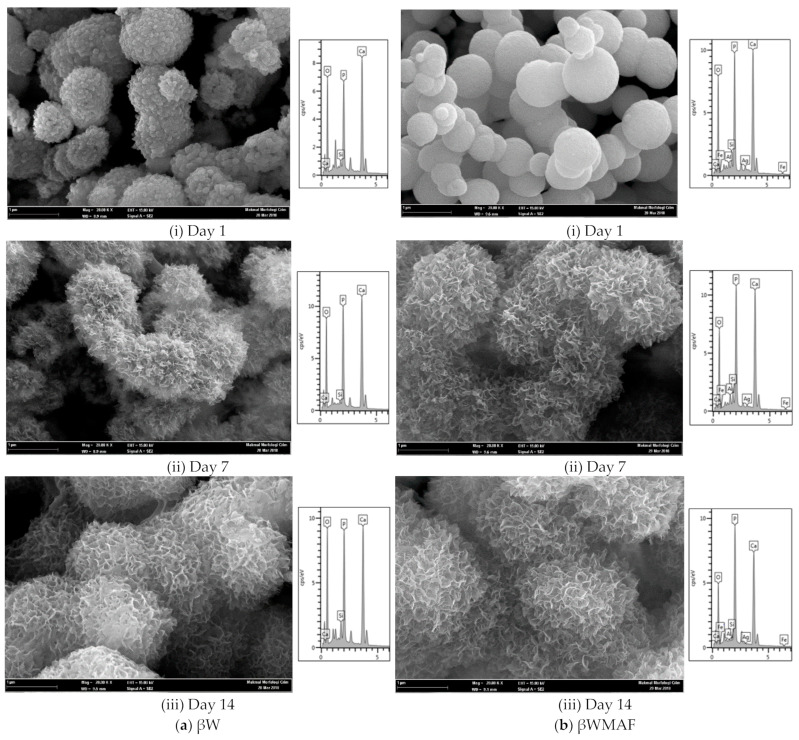
Morphology and elemental analysis of the (**a**) βW and (**b**) βWMAF disc sample after soaking in SBF from day 1 until day 14.

**Figure 7 materials-14-04643-f007:**
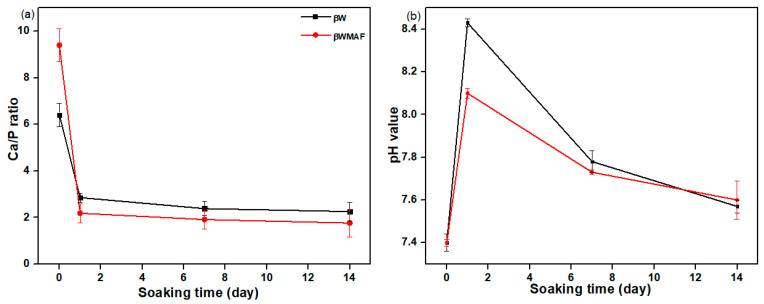
The (**a**) Ca/P ratio and (**b**) pH value of βW and βWMAF disc sample before and after soaking in the SBF up to 14 days.

**Figure 8 materials-14-04643-f008:**
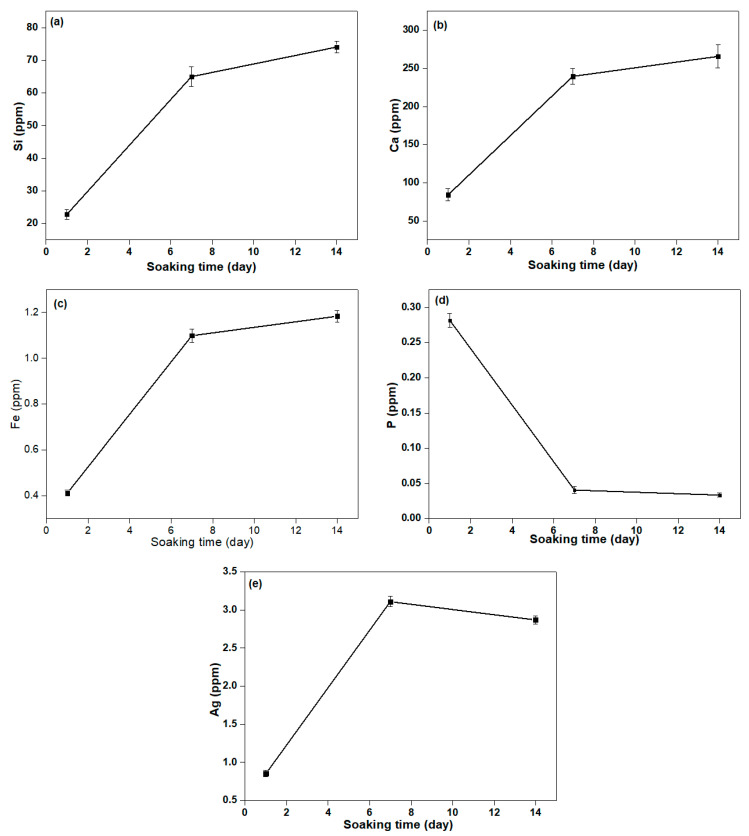
The concentration of (**a**) Si, (**b**) Ca, (**c**) Fe, (**d**) P, and (**e**) Ag ion release from βWMAF samples in the SBF solution for 1, 7, and 14 days.

**Figure 9 materials-14-04643-f009:**
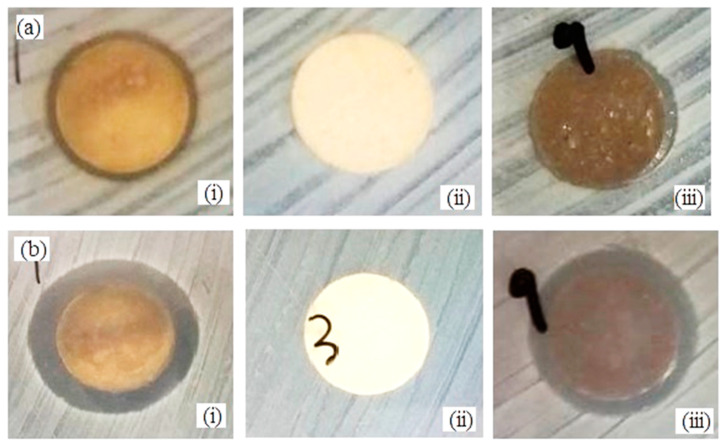
Antimicrobial activity by disc diffusion test of (i) Ag (control sample), (ii) βW and (iii) βWMAF disc for (**a**) *E.coli* and (**b**) *S. aureus*.

**Figure 10 materials-14-04643-f010:**
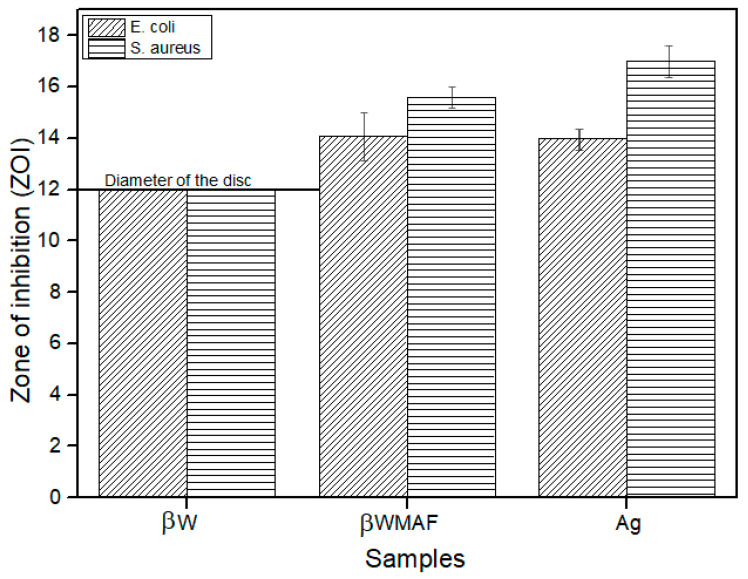
Zone of inhibition (ZOI) of the βW (negative control), βWMAF, and Ag (positive control) disc samples after 24 h.

## Data Availability

Not applicable.

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
