# Peer review of "Effects of Mullite, Maghemite, and Silver Nanoparticles Incorporated in β-Wollastonite on Tensile Strength, Magnetism, Bioactivity, and Antimicrobial Activity"

_materials, 2021, doi:10.3390/ma14164643_

Round 1
Reviewer 1 Report
1. In the introduction part: as the anthor mentioned that ″Still, the cytotoxicity studies of magnetic nanoparticles are restricted because the cytology mechanism persisted not yet investigated″. Accurately, there are many literatures have reported about the cytotoxicity of magnetic nanoparticles, such as the literature ″ Kesse X , Adam A , Begin-Colin S , et al. Elaboration of Superparamagnetic and Bioactive Multicore-Shell Nanoparticles (γ-Fe2O3@SiO2-CaO):A Promising Material for Bone Cancer Treatment. ACS Applied Materials & Interfaces, 2020, 12(42):47820-47830.
- In 2.5 part ,″βW and βWMAF discs were soaked in 30 mℓ of SBF solution for 1, 7, and 14 days in a polyethylene bottle and retain in an incubator at 37℃″. What is the reference? Generally, in vitro bioactivity assay, there are relevant standards or references.
- As bone repair materials, the machanical strenghth including compresive, tensile and bending strength are very important. In 3.2 part, the addition of 20% mullite as reinforcement to the βWMAF increased the tensiles trength to 2.39±0.25 MPa. Whether it matches the strength of the repaire site, in cortical bone or cancellous bone? There was no instructions in this work.
- After 14 days, the released Ag+ concentration was 2.87 ppm, which was higher than that of the reported literations showed in the inroduction part, and the innitial concentration of Ag+ was not stated clearly in this research.
- As you said, mullite as reinforcement for increasing the tensile strength, Ag is help to increase the antibacterial activity of the composite.While at last, the optimial ratio to the comprehensive performance was not dicussed.
- For biomaterial, the in vitro cytotoxicity assay is necessary.
Author Response
Dear Editor,
Here, I attached the review report.
Thank you

Reviewer 2 Report
The authors present the results of a study to produce β-Wallastonite composites for bone-replacement/scaffold applications. Composites of βW with mullite, maghemite and silver are synthesized by dry-mixing the constituents and pressing to form disks of D=12 mm and t=2 mm. The tensile strength, magnetic properties, bioactivity and anti-bacterial potential of these composites are then evaluated.
The problem I have with this study is that the authors have not included any figures in support of their findings. It is therefore impossible for me to assess the methodological and technical soundness of their results.
Some specific points: Ball-mill mixing of macro- and nano-powders (in this case nano-silver was used) is problematic. What is the form/microstructure of the resulting composites? In line 120-123 it is said that this was investigatesd very thoroughly, but in the absence of figures, this cannot be assessed by this reviewer. How uniform was the dispersion of Ag? How homogeneous the surface and most importantly, how homogeneous the sample? How compact were the disks? (extent of internal porosity is unreported). These must be properly documented before the manuscript is considered.
Specifically on the use of the diametral tensile strength, what was the form of the specimens after fracture? Was the fracture truly along the diagonal, or were there secondary cracks and what was their extent? What provisions were made for the contact surfaces? These are not addressed in the manuscript and therefore this segment is inadequate for publication.
What is the signifficance of having a βW composite with magnetic properties?
The anti-microbial and bio-activity studies, along with study of the leaching of ions in the simulated bio-fluid appear complete, but without figures, it is impossible to assess. In any event, it is rather expected that the presence of Ag will indeed result in antimicrobial activity; a question is why do you need to disperse such an expensive form of silver (nano powder) with specific handling requirements, at such high persent (5% w/w) throughout the volume of the composite in order to achieve an effect that is confined to the surface of the part.
My recommendation is resubmit following revision, and to include suitable figures.
Additionally to the diametral test which needs to be supported by actual evidence, the authors need to comment on the apparent difference between the surface appearance of βW and its composite (Fig. 2), the latter also appearing non-homogeneous. What is the overall coherence of the composite samples?
I also notice that the morphologies shown in Figs. 2 and 6 are very different, so much so as to look like different materials (the scale
bars are the same in both cases). Some explanation is needed.
Figure 9 should be made clearer, by showing dimensions and indicating
which area is bacteria free and which is not.
Finally the signifficance of the results shown in Fig 10 needs to be
put in perspective. βW appears to have almost the ZOI of silver, however in the text the authors state "However, no inhibition zones were found on βW, implying that βW lacked bactericidal activity" (lines 361-62). Also the sentence in l. 359-360 does not make sense to me. In general, section 3.6 needs to be better argued in light of
figs. 9 and 10.
English needs to be looked at and some peculiar expressions such as "theatrical release" l. 334, need to be corrected.
Author Response
Dear reviewer,
As per attached.
Thank you

Reviewer 3 Report
- The antimicrobial investigation was carried out using microbes such as Escherichia coli (E. coli; gram-negative) and Staphylococcus aureus using a disc diffusion tecnique (S. aureus; gram-positive)- USUALLY THERE IS THE NEED TO ADD the collection number of the strain used.
- The importance of magnetic properties is not evident(at least to me!) and the authors are kinldky recquested to argue the importnce of these properties.
- 359-360 Several studies showed that S. aureus possesses more vigorous antibacterial activities than E. coli [50].-I DO NOT UNDERSTAND THIS SENTENCE! OR THE AUTHORS WANT TO STRESS THAT-in general- S. aureus is more resistant to antimicrobials – in general- than E coli?
- 361-362 / However, no inhibition zones were found on βW, implying that βW lacked bactericidal activity. CORRECT However, no inhibition zones were found on βW, implying that βW lacked bactericidal OR BACTERIOSTATIC activity.
for th etim ebeing, th emain comment is on fig 9 Fig. 9
Antimicrobial activity by disc diffusion test of (i) Ag (control
sample), (ii) βW and (iii) βWMAF disc for (a) E.coli and (b) S.
aureus. This figure (9) is a) very unclear photographically speaking
! b) I do not understand its title- (i) Ag (control sample), (ii) βW
and (iii) βWMAF/ i sample contains Ag or is a control without any
added chemical/structure?!!!!! disc
the fig 10, the effect of added/tested materials on bacteria( agin there is the need of strain number!!!!) SHOULD BE COMPARED WITH THE EFFECT OF CLASSICAL ANTIBIOTICS WHICH ARE ACTIVE(positive controls) AGAINST THOSE STRAINS (ans antibiotics which are not active against those strains- negative control). without positive controls and negative controls- th emanuscript SHOULD NOT BE ACCEPTED!!!!
Author Response

(The authors gave the same response as above.)

Round 2
Reviewer 1 Report
I have read the revised version of manuscript. The suggestions were inserted in the text. The quality manuscript was improved. Now, my suggestion is that this manuscript can be accepted to be published in this Journal.
Author Response
Dear Prof,
Thank you for your consideration.
Thank you.

Reviewer 2 Report
In response to the author revision/comments (going by the numbers in their response file)
Points3 & 7: The authors have not provided any additional information on the coherence of their samples and details on the diametral testing. The generalities added in lines 141-145 are not actual data. Therefore I consider section 2.3 and any reference to mechanical properties unsuitable for publication.
Point 8: The authors do not explain why the microstructures shown in figure 2, and those in figure 6 that correspond to unsoaked material (top two figures), look so different. In figure 6 the size of the grains is close to 1 micron, while in fig2 it is much smaller. The paper cannot be published with such a glaring discrepancy.
Error bars are missing from Figs 7 &8 .
The paper can be published only after these issues are satisfactorily addressed.
Author Response
Dear Prof,
Thank you for your kindness and attention to read my manuscript.
All the comment as per attached.
Thank you.
